# Improving the Nutritional, Structural, and Sensory Properties of Gluten-Free Bread with Different Species of Microalgae

**DOI:** 10.3390/foods11030397

**Published:** 2022-01-29

**Authors:** Muhammad Waqas Qazi, Inês Gonçalves de Sousa, Maria Cristiana Nunes, Anabela Raymundo

**Affiliations:** 1Department of Food and Health Nofima, Norwegian Institute for Food, Fisheries and Aquaculture Research, Osloveien 1, 1431 Ås, Norway; 2LEAF—Linking Landscape, Environment, Agriculture and Food, Instituto Superior de Agronomia, Universidade de Lisboa, Tapada da Ajuda, 1349-017 Lisboa, Portugal; inesg.sousa@outlook.pt (I.G.d.S.); crnunes@gmail.com (M.C.N.); anabraymundo@isa.ulisboa.pt (A.R.)

**Keywords:** rheology, gluten-free bread, Tetraselmis chuii, Chlorella vulgaris, Nannochloropsis gaditana

## Abstract

Microalgae are an enormous source of nutrients that can be utilized to enrich common food of inherently low nutritional value, such as gluten-free (GF) bread. Addition of the algae species: *Tetraselmis chuii* (Tc), *Chlorella vulgaris* (Cv), and *Nannochloropsis gaditana* (Ng) biomass led to a significant increase in proteins, lipids, minerals (Ca, Mg, K, P, S, Fe, Cu, Zn, Mn), and antioxidant activity. Although, a compromise on dough rheology and consequential sensory properties was observed. To address this, ethanol treatment of the biomass was necessary to eliminate pigments and odor compounds, which resulted in the bread receiving a similar score as the control during sensory trials. Ethanol treatment also resulted in increased dough strength depicted by creep/recovery tests. Due to the stronger dough structure, more air bubbles were trapped in the dough resulting in softer breads (23–65%) of high volume (12–27%) vs. the native algae biomass bread. Breads baked with Ng and Cv resulted in higher protein-enrichment than the Tc, while Tc enrichment led to an elevated mineral content, especially the Ca, which was six times higher than the other algae species. Overall, Ng, in combination with ethanol treatment, yielded a highly nutritious bread of improved technological and sensory properties, indicating that this species might be a candidate for functional GF bread development.

## 1. Introduction

Wheat flour is unique among cereals with the ability to form a viscoelastic network of gluten proteins during mixing with water [1]. The gluten network enables the dough to retain gas during fermentation [2]. The resultant bread product acquires volume and the characteristic fluffy foam-like structure [3]. People suffering from celiac disease are sensitive to gluten or similar proteins [4]. Celiac disease is an autoimmune disorder characterized by the malabsorption of common cereal ingredients, such as gluten [5]. Celiac disease affects approximately 1–2% of the world population, and is still an underestimated disease [6]. To date, the only management strategy is to eliminate gluten from the diet. The European union legislates an upper limit of 20 ppm gluten content on ingredients naturally devoid of gluten. Furthermore, the number of consumers who have gluten-related disorders (wheat allergy, non-celiac gluten sensitivity) is increasing. There are also many consumers who are not diagnosed with these diseases but are eliminating gluten from the diet due to various reasons.

The dough produced without gluten presents poor viscoelastic properties that result in low-quality bread [5]. Various formulations, such as the combinations of gluten-Free (GF) flours (corn, rice, sorghum), starches (cassava or potato) [5,7,8], pseudo-cereals (buckwheat, quinoa), along with hydrocolloids, are used to improve the viscoelasticity of the dough and the subsequent baking quality [7]. Besides poor sensory properties, the GF bread often results in low protein content [8] and may have low micronutrients due to the inherent low nutritional value of the raw materials used [9]. 

Innovative and sustainable ingredients, such as microalgae, possess rich reserves of nutrients (proteins, minerals, vitamins, lipids, antioxidants, and dietary fibers) [10]. Researchers have used microalgae as a functional ingredient in GF breads in the past [8,11,12]. The European Food Safety Authority (EFSA) legislates the commercialization of algae-based food under the Novel Food Regulation (EU) 2017/2470, which defines the novel food as the food not consumed to any significant extent prior to May 1997 [13]. Due to its long history of consumption, *Chlorella vulgaris* and *Spirulina platensis* are recognized as food exempt from novel food application approval. *C. vulgaris* is a green microalga that is cultivated and consumed all over the world [13]. *C. vulgaris* is a well-established source of proteins, providing up to 60% of the needed protein on a dry cell basis. *C. vulgaris* has an amino acid profile comparable to recommended standards by WHO. It is also a source of many other nutrients such as vitamins, carotenoids [14], lipids (EPA, DHA), and antioxidants [15,16]. *Tetraselmis chuii* is approved as a novel food; however, its use is restricted to sauces/condiments with a maximum permissible dose of 250 mg/serving/d [17]. *T. chuii* is considered a complete source of proteins providing all the essential amino acids comparable to the WHO/FAO reference profile [18]. Besides proteins, *T. chuii* has high levels of minerals—~16% on a dry weight (DW) basis—namely calcium, phosphorus, and sulfur [19]. Among the algae species with a status of pending approval [20], *Nannochloropsis gaditana* (also named *Microchloropsis gaditana*) presents enormous potential as a healthy novel food ingredient. It accumulates a high amount of quality proteins up to 47% DW [21], with an essential amino acid index score of EAAI = 1.02, which is higher compared to the approved species of *Spirulina* (EAAI = 0.81) and *Chlorella* (EAAI = 0.92) [22]. *N. gaditana* also has a very high lipid content of ~22%, with 3–6% PUFA, and up to 0.6 mg/g DW carotenoids. *N. gaditana* also possesses rich reserves of vitamins and minerals [20]. 

One of the most common problems noticed when microalgae are used as a functional ingredient in bread is the severe deterioration of the sensory properties (dark green color, flavor, and odor). Microalgae acquire their dark green color from pigments (chlorophylls). These attributes can contribute to the decreased acceptance of enriched bakery products and limit the incorporation levels [23]. 

In the current study, the algae biomass was bleached using ethanol extraction. Three microalgae species: *Tetraselmis chuii* (TcR), *Chlorella vulgaris* (CvR), and *Nannochloropsis gaditana* (NgR), and the corresponding ethanol-treated biomasses, *Tetraselmis chuii*-treated (TcT), *Chlorella vulgaris*-treated (CvT), and *Nannochloropsis gaditana*-treated (NgT), were replaced in the GF bread recipe to study their impact on dough rheology, baking performance, and sensory attributes. This was coupled with a nutritional analysis of GF-breads, while an attempt was made to evaluate if empirical (farinograph) and fundamental (G’/G” and creep/recovery) rheology could predict the baking performance of the GF breads comprised of different nutritional compositions. Therefore, the ultimate goal was to develop a highly nutritional GF-bread enriched with microalgae without a major compromise on the technological and sensory attributes.

## 2. Materials and Methods

### 2.1. Raw Materials and Preparation of Doughs

GF breads were developed according to a previously optimized formulation [12]. The control and 4% (*w/w*, total flour basis) microalgae formulations are summarized in Table 1. The thickener, hydroxypropyl methylcellulose (HPMC), was kindly supplied by Dupont. HPMC contributes to increased dough viscosity, with a positive impact on volume and bread texture. Flours and other ingredients were purchased from local markets, and distilled water was used. The preparation of dough and baking of bread were conducted according to the previously optimized method [12]. At least three loaves of each formulation were prepared, and all analyses were performed a minimum of three times.

The three species of microalgae (TcR, CvR, NgR) were produced in the A2F (Algae to Future) project. The ethanol extraction of the given three species was performed at Nofima, Norwegian Institute for Food, Fisheries, and Aquaculture Research, Ås, Norway, as published in detail elsewhere [24]. The macronutrient composition reported for the biomass was kindly provided by the A2F biomass suppliers and is published in [24].

### 2.2. Farinograph Mixing Properties

Farinograph mixing properties were estimated using the Micro-doughLab 2800 (Perten Instruments, Sidney, Australia). The formulation was adjusted according to the 4 g bowl and 14% moisture basis. For moisture determination of the ingredients, an automatic moisture analyzer PMB 202 (Adam Equipment, Oxford, NJ, USA) was used, and the weights of flours were corrected according to their moisture content. Mixing in the farinograph was performed following the protocol “General Flour Testing Method”, i.e., mixing speed 63 rpm at 30 °C for 20 min. A peak torque value of ~69 mN.m was recognized as optimum for better baking properties of the control GF bread, after the preliminary experiments. The rest of the algae formulations were kept in this range of the torque with a 5% cut-off by modifying the water addition. From the mixing curves, peak resistance (mN.m), dough development time (DDT), stability (DS), softening (mN.m), and additionally, the mixing torque recorded at 600 s (T600) and 1200 s (T1200) are reported. 

### 2.3. Fundamental Rheology-SAOS and Creep Recovery

Small amplitude oscillatory shear (SAOS) and creep/recovery measurements were performed on a Haake MARS III (Thermo Fisher Scientific, Waltham, MA, USA), equipped with a UTC Peltier. A serrated plate-plate geometry with a 20 mm diameter was used in the measurement at 20 ± 0.5 °C. Dough slices (made in farinograph) cut out from the middle of the dough balls were placed on the lower plate and compressed at a speed of 0.6 mm/min between the two plates to the set gap of 1 mm. Excess dough was trimmed from the edges, and paraffin oil was added around the sample to prevent moisture loss during measurements. The sample was rested for 5 min, after which the frequency sweep at 0.1–100 Hz was started within the linear viscoelastic region (LVER), 10 Pa (previously determined by using stress sweeps). The acquired data were fitted to the power-law model: G’(ω) = K’ (ω n’), and G” (ω) = K” (ω n”)(1)
where G’ is the storage modulus (Pa); G” is the loss modulus (Pa); ω is the angular frequency (rad/s); and K’, K” (Pa.sn), n’, and n” are the power-law parameters. Immediately after the frequency sweep, a time sweep was conducted at a fixed frequency of 1 Hz, and a stress of 10 Pa for 10 min was applied. No change in the G* and G** confirmed a relaxed sample prior to the creep/recovery test. During the creep test, the sample was subjected to a constant stress of 100 Pa for 2 min (outside the LVER). After removal of the stress, the samples recovered for 6 min. The stress and duration of its application were chosen based on initial experiments to achieve the steady state. The data from the creep test may be presented by the creep compliance function (J): J = f(t) = γ/σ (1/Pa)(2)
where γ is the strain, σ is the constant stress applied during creep/recovery experiments. For the creep/recovery test, the parameters: maximum creep compliance (Jmax), elastic recovery compliance (Je), viscous recovery compliance (Jv), and zero shear viscosity(η_0_) were used in the current study.

### 2.4. Bread Technological Properties (Color, Volume, and Firmness)

Bread crust/crumb color was measured using a Minolta CR-400 (Japan) colorimeter. The results are presented following the CIELAB system: L*—lightness (0 black to 100 white); a* green to red (−60 to 60), and b* blue to yellow (−60 to 60). The total color difference (∆E) was estimated from the equation:(3)$$\Delta E=\sqrt{{\left(\Delta L^{*}\right)}^{2}+{\left(\Delta a^{*}\right)}^{2}+{\left(\Delta b^{*}\right)}^{2}}$$

Each bread was measured 6 times, while the values presented are an average of 3 independent baking trials. Bread volume (n = 3) was estimated following the rapeseed displacement method AACC 10-05.01. Bread crumb texture was measured 2 h after baking (at 20 °C), using a texture analyzer TA.XTplus (Stable MicroSystems, Surrey, UK) equipped with a 5 kg load cell and cylindrical probe diameter of 10 mm that was allowed to penetrate into the manually sliced ~20 mm slice of the bread. The maximum resistance to penetration was presented as the firmness (N). Measurements were made twice on 3 slices of the same loaf (n = 6). The results presented are an average of 3 independent baking trials. 

### 2.5. Bread Nutritional Properties

The moisture content of bread was measured gravimetrically through an automatic moisture analyzer PMB 202 (Adam Equipment, Oxford, NJ, USA) at 130 °C to a constant weight. The rest of the nutrients and bioactive compounds were measured in dried breads (<3% moisture) ground to powdered form. The total ash content was determined by incineration at 500 °C in a muffle furnace (AACC 08–01). Protein content (N× 6.25) was estimated by the combustion method. DUMAS [25], using a Vario EL elemental analyzer (Elementar, Langenselbold, Germany). The carbohydrate content was calculated by the difference between the protein, lipid, ash, and moisture contents of the breads. Minerals (K, Ca, Mg, P, S, Fe, Cu, Zn, and Mn) were estimated using an Inductively Coupled Plasma Optical-Emission Spectrometry (ICP-OES) Thermo-Scientific^TM^ iCap Series-7000 (Thermo-Fisher-Scientific, Waltham, MA, USA), following the method described by [26]. Fat-content was determined following the Portuguese standard method NP4168 (NP 4168., 1991) described by [27]. All the analyses were performed at least in duplicate.

### 2.6. Evaluation of Bioactivity in Breads

To estimate the total phenolic compound content, extraction was performed by dissolving 2 g/10 mL dried bread powder in 96% ethanol and centrifuged at 7000× *g* rpm for 10 min. The samples were filtered, and the solvent was evaporated under vacuum in a rotatory evaporator. The dried extracts were dissolved in dimethyl sulfoxide (DMSO) to obtain 20 mg/mL stock solution and stored at +4 °C until the experiments were conducted. The total phenolic content (TPC) of bread extract was evaluated using the method adapted from Mohankumar [28]. Slight modifications are detailed elsewhere [12].

The scavenging effect of bread extracts was determined using the DPPH(2,2-diphenyl-1-picryl-hydrazyl-hydrate) methodology [29] as detailed elsewhere [12].

Determination of the total pigments was conducted by adding 3.8 mL of ethanol to 200 μL of extract. The absorbance was measured at 470, 648, and 664 nm corresponding to carotenoids, chlorophyll a (Chl-a), and chlorophyll b (Chl-b), respectively. Ethanol was used as a blank. The values were determined using the following equations [30]:Chl-a = 13.36 × A_664_ − 5.19 × A_648_(4)
Chl-b = 27.43 × A_648_ − 8.12 × A_664_(5)
Carotenoids = (1000 × A_470_ − 1.63 × Chla − 104.96 × Chlb)/221(6)

### 2.7. Sensory Evaluation

Sensory analysis was conducted in a standardized test sensory room with booths, following standard EN ISO 8589: 2007 procedure. An untrained panel (n = 33; gender: females 19, males 14; age range: 19–73) participated in the hedonic evaluation following the commonly used protocol by LEAF [11,12,26] in accordance with the ethical standards of the local committee responsible for human experiments and with the code of ethics of the World Medical Association (Declaration of Helsinki 1975, revised in 2013). Samples were randomly distributed, and the panelists were invited to sufficiently cleanse their palate with apples between samples. Besides the control, only the breads with *Chlorella vulgaris* (CvR, CvT) were offered to the panel since it is the only species currently approved by the EFSA. The panelists judged the bread color, smell, aroma, flavor, texture, and global appreciation on a 5-point hedonic scale from very pleasant (5) to very unpleasant (1). Buying intent was also assessed on a 5-point scale from: would always buy (5) to would never buy (1).

### 2.8. Statistical Analysis 

Analysis of variance (ANOVA) was conducted using a general linear model (GLM) in Minitab 19 software. Prior to ANOVA, and where necessary, a Box-Cox transformation was applied to all raw data to achieve a near-normal distribution. Equal variance was confirmed by conducting a test for equal variances. The pair-wise comparison presented in figures/tables was acquired by a Fisher LSD test at a 95% confidence interval. Principal component analysis (PCA) was conducted using Unscrambler software (Version10.3 A/S Trondheim Norway). To find the correlations among composition, rheology, and baking, Pearson correlation was performed using Microsoft Excel.

## 3. Results and Discussion

### 3.1. Impact of Algal Biomass Addition on Dough Rheology

#### 3.1.1. Empirical Methods

Representative mixing curves acquired with Micro-doughLab and the resulting mixing properties are presented in Figure 1 and Table 2. Generally, with microalgae biomass substitution, a slight increase in the water absorption (WA) was needed to achieve the target peak torque value of the control, except for the TcR. From the mixing curve, the mixing tolerance followed over 20 min (standard method) was different for the control and for the microalgae biomass-replaced doughs. The doughs made with TcT, CvT, and NgT presented a more stable torque over time. To understand this better, the mixing torques T_600_ and T_1200_ were followed and reported here. The T_600_ and T_1200_ were significantly higher (*p* < 0.05) for the TcT and NgT compared to the corresponding untreated microalgae TcR and NgR. Only the CvT compared to the CvR at T_1200_ showed the opposite behavior. Different doughs showed different degrees of dough softening; the TcR-replaced dough presented the highest dough softening, followed by the control. The ethanol-treated algae biomasses (TcT, CvT, and NgT) seemed to result in more stiff doughs compared to the corresponding raw biomasses, TcR, CvR, and NgR forms. This effect was significant between the TcR and corresponding TcT-replaced doughs. Differences in stability (DS) were not significant in any of the given combinations of doughs. The doughs made with algal biomass replacements required slightly higher WA and took a longer (DDT) time to develop compared to the control. 

#### 3.1.2. Fundamental Rheology-SOAS and Creep Recovery

Figure 2 shows typical creep recovery curves and mechanical spectra’s while Table 3 presents different parameters from the creep/recovery test and the G’ and G” during SAOS frequency sweep tests. The control GF dough showed the highest Jmax, Je, and Jv, followed by the CvT. In general, the treated doughs (TcT, CvT, and NgT) showed higher Jmax, Je, and Jv compared to corresponding raw biomass-replaced doughs (TcR, CvR, and NgR). The Jmax for the control was significantly higher between the control and TcR/TcT, and NgR and NgT; however, for Je, which is a direct estimate of the dough elasticity, a significant difference between the control and two out of three raw biomasses TcR and NgR were noticed. Only the NgT among the treated algae biomasses yielded a significantly lower Je compared to the control. Similarly, the η_0_, which is an indicator of the flowability of a material, and is oppositely related with the Jmax/Je [31], demonstrated lower values for the control and for all the ethanol-treated (TcT, CvT, and NgT) samples compared to the corresponding raw (TcR, CvR, and NgR) biomass-replaced doughs. The highest η_0_ for the dough made with TcR followed the NgR. Only one instance, between the control and CvR, was a significant difference noticed for η_0_, indicating that these two are very different samples.

The results from the frequency sweep showed an increase in both G’ and G”, with G’ > G” for all samples. Compared to the control, higher G’ and G” values were recorded in the algae biomass-replaced doughs (TcR, TcT, NgR, and NgT); however, the CvR and the corresponding CvT presented lower G’ and G” compared to the control. The influence of ethanol treatment on G’ was not clear; the biomasses in the case of CvT and NgT in the G’ increased compared to the corresponding CvR and NgR, while the opposite was noticed between the TcR and the TcT-replaced GF doughs. The power law-indices n’ and n” for the G’ and G” showed a similar increase for the control and the rest of the composition with a median of 0.22 for n’ and 0.21 for n”, respectively. To elucidate the impact on parameters G’ and G”, by the GF dough and different algae biomasses, multivariate statistics were performed (discussed later).

#### 3.1.3. Nutritional and Technological Properties of the Bread 

GF bread, which normally has low nutritional value, recorded a general increase in the protein, fat, and ash content by the incorporation of the algal biomasses (Table 4). The protein content significantly increased with the replacement of 4% of all algal biomasses due to the high protein content of the algae biomass compared to the other flours. Moreover, higher protein enrichment was recorded with the treated than with the raw biomasses. The increase in protein content was significant (*p* < 0.05) with CvT and NgT-replaced samples compared to the corresponding CvR and NgR-replaced biomasses. The lipid content recorded higher values in the raw biomasses (TcR, CvR, and NgR) compared to the control. No difference in the lipid content was noticed in the breads baked with TcT, CvT, and the control, due to the ethanol treatment removing most of the lipids from the treated biomasses. Similarly, the ash content recorded a significant increase compared to the control in most cases due to the inherent high ash content from the substituted algae biomass, as reported earlier (Table 1). The moisture content in the baked breads and the carbohydrates were similar in all formulations, while in general, the calorie intake slightly increased in the alga breads compared to the control due to the high lipid content in TcR, CvR, and NgR and high protein content in the TcT, CvT, and NgT biomasses. The increase in the nutritional value of GF bread resulting from the algal biomass addition was consistent with previous studies [8,11,12] using ~4% algal biomass incorporation. Common in all previous studies is that the increase in nutritional value is at the expense of deteriorating sensory appeal observed in consumer tests. 

The mineral profile improved in all the GF breads enriched with algae biomass, compared with the control bread (Table 5). Breads with TcR and TcT were particularly high in Ca and Fe. Meanwhile, K, Mg, Cu, Zn, S, and Mn increased to a similar extent, regardless of the type of algae biomass used. The recommended daily value (RDV) status was possible to achieve only in the case of Mg, P, Fe, and Mn in all combinations of bread, including the control. 

Cereals are generally high in minerals, except in Ca [32], and therefore, RDV requirements for several minerals have been fulfilled, even by the control. With the algal biomass enrichment, an improvement of the mineral content was achieved. Using a similar formulation of GF bread [11,26] obtained generally higher values, even for the control, by enriching formulations with acorn flour and microalgae, respectively. Among the flours used, buckwheat utilized the highest amount in the GF formulation, which is especially rich in Mg, P, S [26]. Ca content was significantly increased by the *Tetraselmis*, indicating that the species provides an alternative in situations where bread with high Ca content is desired. The high Ca content has been identified as a special trait of this species, acquired through a complex biomineralization process discussed in detail elsewhere [33].

Replacement of the GF breads with 4% microalgae demonstrated an increase in the bread firmness and a decrease in its volume vs. the control (Figure 3), with breads becoming more compact. This confirms that at low contents, raw microalgal protein induces a destabilization of the network formed by starch and HPMC [12]. The bread prepared with TcR and CvR biomasses was significantly firmer vs. the control GF breads. Ethanol-treated biomasses (TcT, CvT, and NgR) generally led to a decrease in crumb firmness vs. the corresponding raw biomasses (TcR, CvR, and NgR). This effect was significant in the bread made with CvT vs. the corresponding CvR. Similarly, the specific volume of the control bread was significantly higher than the rest of the compositions. The ethanol treatment significantly increased the bread volume in the case of TcT compared to the TcR counterpart, while an increased bread volume for the other combination (CvT and NgT) was witnessed vs. the corresponding CvR and NgR-replaced algae biomass breads.

#### 3.1.4. Combined Discussion on Biomass Nutritional Composition, Dough Rheology, and Consequential Baking Properties 

PCA analysis was performed to establish any possible correlation between dough rheology and subsequent properties of the bread with different nutritional compositions (Figure 4). The figure shows three distinct patterns; raw algae biomasses (TcR, CvR, and NgR) were relatively closer and within the same quadrant. These samples were firm with high lipid and ash contents, which probably does not allow the proteins to express themselves and provide structure to the dough and subsequently to the breads. The doughs and breads with control and CvT were close to each other, demonstrating a high volume, and they were described best by the creep parameters (Jmax, Je, and Jv). Pearson correlation showed a strong relationship between the volume and Je (r = 0.90), bread volume, and Jmax (r = 0.92), respectively. Wang and Sun [33] also found a strong correlation between Je and the bread volume using wheat flour. Contrarily, a negative correlation between crumb firmness and Je (−0.72) and Jmax (r = −0.77) was recorded.

Unlike the creep/recovery test, the SAOS parameters (G’ and G”) failed to demonstrate any clear relationship between the dough strength and baking properties (bread volume/firmness). Previous studies could not establish any clear relationship between the SAOS and baking performance [1,2,34]. These studies suggest that the very low deformation conditions employed within the linear viscoelastic region (LVER) in SAOS may not be relevant to the larger deformations experienced by the dough during the mixing and baking steps.

Contrarily, deformation outside the LVER (creep/recovery), as employed in creep/recovery tests, seemed to correlate better with the baking performance in the GF breads. The PCA plot indicated that η_0_ predicted the correlation of bread volume (r = −0.76) and firmness (r = 0.80) quite well. The doughs (control, CvT, TcT, NgT) with lower η_0_ values resulted in bread with improved baking properties. η_0_ is an indicator of the ease with which material flows and is extracted from the peak region of the creep phase. Therefore, η_0_ consists of both the viscous and elastic characteristics of a material. Since a combination of both the elastic and viscous properties are required for optimum baking performance [30]), our results suggest that a high Jmax and low η_0_ of the dough might be useful in predicting GF bread volume and firmness.

Higher WA and DDT in the algae doughs were needed due to the presence of more proteins in the given algal biomasses. Proteins usually increase the WA and DDT, as previously noticed [35]. The higher T_600_ and T_1200_ in the treated biomass-based doughs were probably due to the purification of algae proteins by the ethanol treatment previously noted [36]. This argument is strengthened by the dough softening, an estimator of the rate of deterioration of dough strength, after reaching the peak development, which recorded the lowest value for the TcR (less protein content compared to the TcT), followed by the low protein control GF dough. The empirical farinograph method inadequately discriminated the different GF doughs used in the current study since they were developed for wheat flour. Nevertheless, farinograph is a useful tool to standardize doughs with different compositions by adjusting the water addition to acquire the Peak torque. The PCA plot suggested that TcR was not well described by any of the rheological (empirical or fundamental) methods. Ethanol treatment had a profound impact on TcR, leading to a stable dough illustrated by parameters DS, T_600s_, and T_1200s_ and, consequently, resulting in improved technological properties.

The third pattern that the PCA plots elucidated was the enrichment of proteins (more noticeable in the TcT and NgT) that occurred due to ethanol extraction, which seems to strengthen the dough network expressed by high elastic modulus (G’) and higher torque T_600s_ and T_1200s_ by farinograph. The ethanol treatment was intended to eliminate the green pigments that inevitably removed the lipids. Among the lipids, the high unsaturated fatty acids from microalgae [22] have been shown to reduce bread volume [37], probably, by preventing the continuous network formation that is facilitated by the HPMC in a GF dough. The treatment virtually removed all the chlorophylls/phenolic compounds, which further boosted the protein content. The enriched proteins from microalgae (resulted from ethanol treatment) and probably from flours of the GF formulations seemed to interact better with HPMC and with the starch component, promoting a continuous network, that resulted in greater air entrapment during fermentation and increased loaf volume, as discussed in a previous study [38]. A previous study on GF bread enriched with protein from egg and milk, along with rice flour and HPMC, showed the creation of a “bicontinuous matrix with starch” that mimicked the gluten network [39].

Future work is proposed to examine the role of pure algae protein (from different species) in dough strength instead of the whole biomass with respect to a broad range of rheological classification, especially using extensional rheology. 

#### 3.1.5. Bioactivity in Bread (Phenols, Antioxidant Capacity, and Pigments)

Higher TPC values were found in breads with raw microalgae (TcR, CvR, and NgR), as expected (Figure 5). Treatment with ethanol significantly reduced the TPC level in all the corresponding breads with TcT, CvT, and NgT. The control bread had higher TPC than the ethanol-treated algae breads. A significantly higher antioxidant activity was recorded in all of the raw algae breads (TcR, CvR, NgR), compared with the treated counterparts (TcT, CvT, NgT) measured with DPPH. As expected, the greenish breads baked with raw microalgae (TcR, CvR, NgR) were dominated by the chlorophyll-a, while chlorophyll-b and carotenoids were also present, but in lesser amounts. Ethanol treatment completely removed the pigments. Hence, breads baked with TcT, CvT, and NgT were entirely devoid of the pigments, just like the control (shown in Figure 5).

Polyphenols possess antioxidant activity [40]; therefore, a strong positive correlation in phenolic content and antioxidant activity (r = 0.99) was noted. A high TPC and antioxidant activity noticed in the control was probably due to the presence of buckwheat in the GF recipe in the dominant quantity. Buckwheat has previously been shown to possess high polyphenols and antioxidant activity in bread [41]. 

The microalgae biomass in its raw form had a dark green color due to the pigments, which were completely removed by the ethanol treatment. A previous study [42] concluded that ethanol was the most efficient solvent for chlorophyll extraction. Moreover, as the *Nannochloropsis* in its native form is completely devoid of chlorophyll-b, as explained previously [43], the lack of chlorophyll-b in NgR was not a surprise.

#### 3.1.6. Bread Color and Sensorial Evaluation 

The images in Figure 6 clearly show differences in GF bread crust/crumb color that was perceivable by the human eye, as confirmed instrumentally, i.e., ∆E > 3 [44]. In terms of crumb brightness L*, the treated biomasses TcT, CvT, and NgT showed significantly higher L* vs. the corresponding TcR, CvR, and NgR-replaced GF breads (Table 6). The NgT demonstrated the most profound influence by the ethanol treatment leading to a non-significant L* compared to the control. The L* for the control vs. the rest of the composition was significantly higher. Ethanol treatment of TcR, CvR, NgR eliminated the green color demonstrated by negative a* converted to positive a* in the corresponding TcT, CvT, and NgT-replaced bread crumbs. Like the L*, the a* for NgT and the control were similar. The degree of yellow (b*) was significantly higher, only between the CvR and the corresponding CvT and the control vs. CvR, while in general, the b* values increased for the TcT and NgT vs. the corresponding TcR and NgT. Against the control, the ∆E* for the crumb recorded a > 50% decrease in the TcT and CvT vs. the corresponding TcR and CvR. Meanwhile, ∆E* decreased more than four times in the NgT vs. the corresponding NgR-replaced breads. Similar trends of bread crust brightness L* were noticed in the crumb, i.e., the control was significantly brighter vs. all the raw biomasses (TcR, CvR, and the NgR), while the L* was like the NgT. Contrarily to the bread crumb, the bread crusts resulting from the TcR, CvR, and NgR were no longer green (positive a*); however, the corresponding TcT, CvT, and NgT still transformed significantly towards the red hue. Like the bread crumb, the a* for the NgT was like the control. The yellowness (b*) degree was highest in the control, which decreased to less extent in the NgT, CvT, and TcT vs. the corresponding NgR, CvR, and TcR replaced breads. Like the crumb, the color difference ∆E* in crust vs. the control recorded a general reduction. Hence, ∆E decreased ~80% in TcT, nearly three and five times in CvT and NgT, respectively, vs. the corresponding TcR, CvR, and NgR crust colors. The crust color with the NgT-based bread (∆E* = 3.9) was no longer perceivable by the naked eye.

Microalgae contain pigments (chlorophylls and carotenoids), which are responsible for their dark green color [23]. The positive a* in treated biomass-based bread, verified the complete removal of pigments consistent with the spectrophotometry results shown earlies. The influence was particularly promising in the *Nannochloropsis,* which is currently not approved as food. The species demonstrated ∆E* = 7.1 for the crumb and ∆E* = 3.9 for the crust, indicating that the bread crust/crumb color was nearly not detectable by the eye. One of the goals with ethanol treatment was to produce a GF bread color close to the control, which was almost achieved with *Nannochloropsis*. 

The CvT-replaced GF bread was evaluated for its sensorial attributes against the corresponding CvR and the control. Figure 7 demonstrated that CvT was appreciated by the panelists, like the control for the attributes general appearance, color, aroma, and flavor. For these attributes, the CvR acquired the lowest scores consistent with the literature [8]. The texture attributes of the three breads acquired similar scores with a slight preference for the CvT (4.06 ± 0.92), followed by the CvR (3.94 ± 0.74), and the control (3.79 ± 0.95).

Overall, the panelists showed a significantly higher “global appreciation” for the CvT and the control vs. CvR-replaced bread. The differences in “global appreciation” for the CvT and the control were non-significant. Similar to the global appreciation, a high “buying intent” was shown for the control and the CvT vs. the CvR-replaced breads. Generally, the results revealed that the ethanol-treated CvT was brought significantly closer to the level of the control for nearly all sensorial attributes. 

Consistent with the color results acquired instrumentally, the CvT was appreciated in sensorial tests like the control due to the elimination of green pigments from the biomass. The aroma and flavor of CvT were appreciated nearly the same as the control compared to the CvR-based bread due to the elimination of volatile compounds. The algae biomass in its native form contains sulfur compounds, which are responsible for the perceived aroma [13]. These compounds are largely eliminated by ethanol treatment, as previously shown [36]. Furthermore, the ethanol treatment strengthens the dough structure allowing more air entrapment, resulting in improved technological properties, as shown early. This increased the general appearance and global appreciation score of the CvT vs. CvR. The panelists perceived the CvT bread as similar to the control and, therefore, showed the same buying intent for CvT as for the control. Overall, the sensorial study showed that ethanol treatment or similar processing steps aimed at bleaching the algal biomass are promising strategies to improve the sensory properties. However, the results from the sensory evaluation must be interpreted with some caution; only 33 untrained panelists had participated. Nevertheless, our results are promising in terms of consumer acceptance.

## 4. Conclusions

Incorporation of the algae biomass into gluten-free bread elevates its protein, lipid, mineral, and antioxidant activity vs. the control with a compromise on the bread volume, texture, color, and sensory attributes. To counter this, the given algae biomasses were treated with ethanol, which eliminated pigments and yielded lighter biomass colors. Ethanol-treated biomass also strengthened the dough, resulting in improved volume, texture, and color. All the ethanol-treated algae-based breads contained higher protein compared to the corresponding raw microalgae-based breads. *Tetraselmis chuii* increased the calcium content of the GF-bread more than the other species, while the most promising improvement in technological properties of bread was noticed in the *Nannochloropsis gaditana* based GF-breads.

## Figures and Tables

**Figure 1 foods-11-00397-f001:**
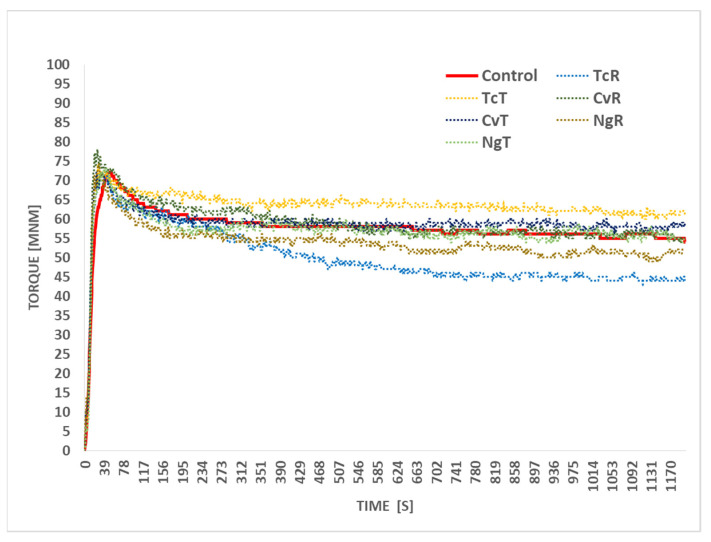
Representative farinograph mixing curves acquired from the gluten-free dough (control) and the ones with 4% replacement of Tetraselmis chuii (TcR), Tetraselmis chuii ethanol-treated (TcT), Chlorella vulgaris (CvR), Chlorella vulgaris ethanol-treated (CvT), Nannochloropsis gaditana (NgR) and Nannochloropsis gaditana ethanol-treated (NgT).

**Figure 2 foods-11-00397-f002:**
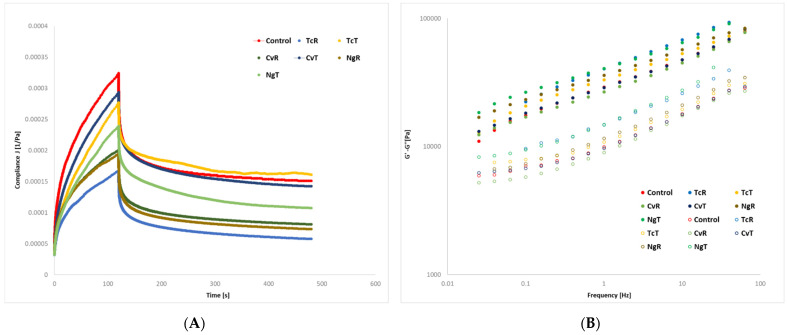
Representative creep and recovery curves (**A**) and mechanical spectra (**B**) of the control dough and those with a 4% replacement of *Tetraselmis chuii* (TcR), *Tetraselmis chuii* ethanol-treated (TcT), *Chlorella vulgaris* (CvR), *Chlorella vulgaris* ethanol-treated (CvT), *Nannochloropsis gaditana* (NgR), and *Nannochloropsis gaditana* ethanol-treated (NgT).

**Figure 3 foods-11-00397-f003:**
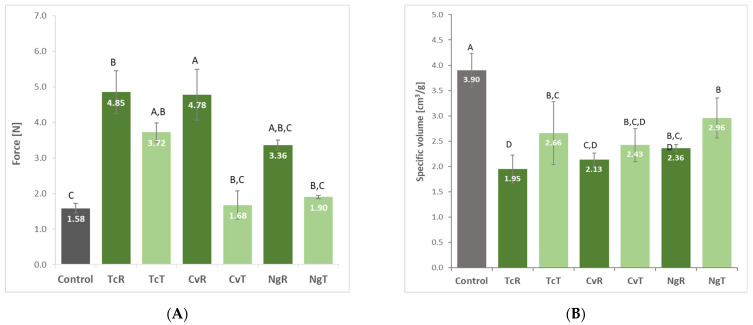
Crumb firmness (**A**) specific volume (**B**) of breads prepared with *Tetraselmis chuii* (TcR), *Tetraselmis chuii* ethanol-treated (TcT), *Chlorella vulgaris* (CvR), *Chlorella vulgaris* ethanol-treated (CvT), *Nannochloropsis gaditana* (NgR), and *Nannochloropsis gaditana* ethanol-treated (NgT). The values represent mean ± stdev of at least 3 replicates. Different letters for a given parameter indicate a significant difference (*p* > 0.05) using the Fisher LSD test.

**Figure 4 foods-11-00397-f004:**
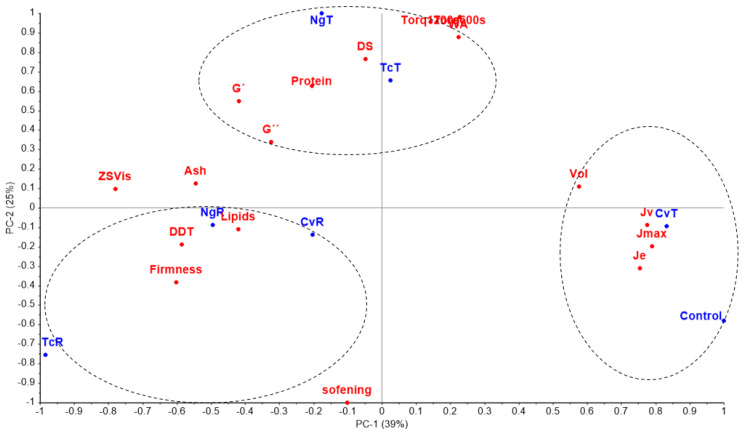
PCA Bi-plot demonstrating relationship among rheology tests: farinograph (WA = water absorption, torque600s, torque1200s, softening is dough softening), oscillatory, (G’ and G”) creep (Jmax, Je, Jv, ZSV is zero shear viscosity), bread (firmness, vol is volume), and nutritional properties for the doughs and breads made with control and *Tetraselmis chuii* (TcR), *Tetraselmis chuii* ethanol-treated (TcT), *Chlorella vulgaris* (CvR), *Chlorella vulgaris* ethanol-treated (CvT), *Nannochloropsis gaditana* (NgR) and *Nannochloropsis gaditana* ethanol-treated (NgT). Blue colour for the samples and red colour refers to measurements parameters.

**Figure 5 foods-11-00397-f005:**
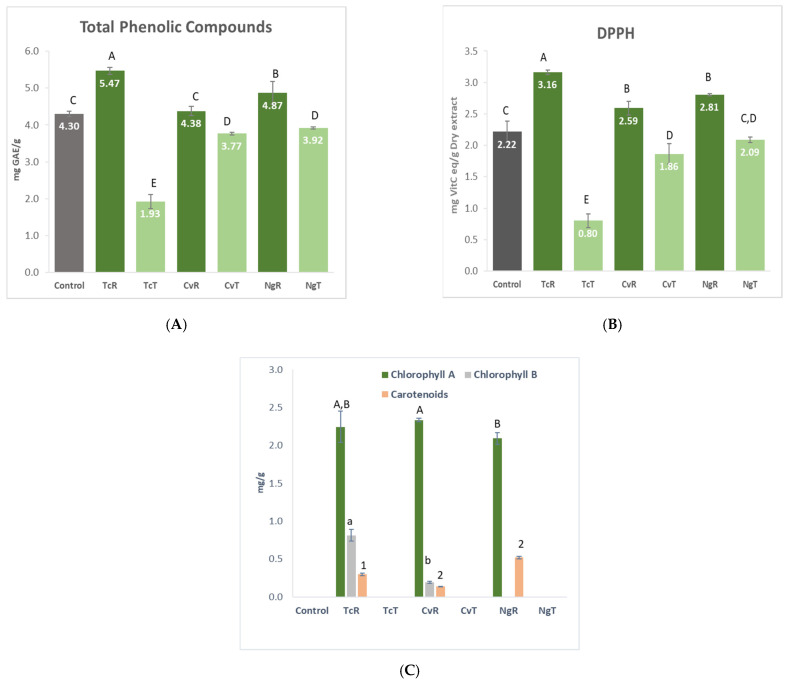
Measurement of the antioxidant capacity using total phenols (**A**), DPPH (**B**), and pigments (**C**) in the given gluten-free breads enriched with *Tetraselmis chuii* (TcR), *Tetraselmis chuii* ethanol-treated (TcT), *Chlorella vulgaris* (CvR), *Chlorella vulgaris* ethanol-treated (CvT), *Nannochloropsis gaditana* (NgR) and *Nannochloropsis gaditana* ethanol-treated (NgT). Totality dark refers to control (**A**,**B**). Values represent mean ± stdev (n = 3). Different letters for a given parameter indicate a significant difference (*p* > 0.05) using a Fisher LSD test. Significant differences in chlorophyll a, chlorophyll b, and carotenoids are shown by capital letters, small letters, and numbers, respectively.

**Figure 6 foods-11-00397-f006:**
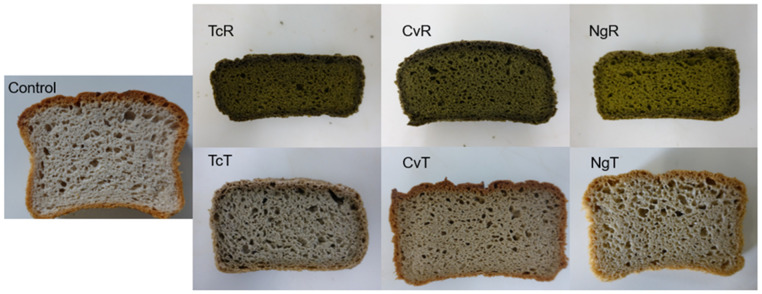
Images of the gluten-free bread with the 4% replacement of microalgae’s: *Tetraselmis chuii* (TcR), *Tetraselmis chuii* ethanol-treated (TcT), *Chlorella vulgaris* (CvR), *Chlorella vulgaris* ethanol-treated (CvT), *Nannochloropsis gaditana* (NgR), and *Nannochloropsis gaditana* ethanol-treated (NgT).

**Figure 7 foods-11-00397-f007:**
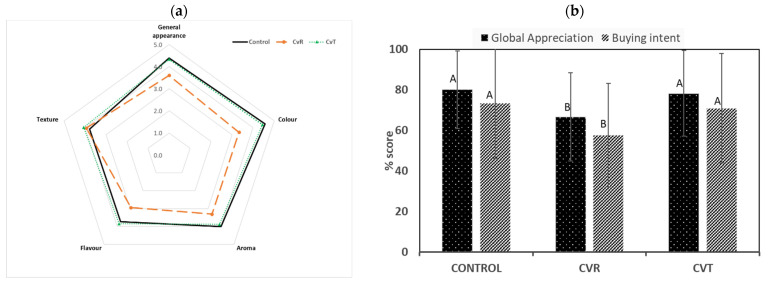
Average responses from the panelists (n = 33) for the control and the *Chlorella vulgaris* raw (CvR) and *Chlorella vulgaris* ethanol-treated (CvT) biomass-replaced (4%) breads for different sensorial attributes (**a**) and for global appreciation and buying intent (**b**). Different letters (where mentioned) indicate a significant difference (*p* > 0.05) using the Fisher LSD test.

**Table 1 foods-11-00397-t001:** Formulation of the breads and macronutrient composition (%) of the raw ingredients used for the control and different 4% microalgae biomass-replaced breads. *Tetraselmis chuii* (TcR), *Tetraselmis chuii* ethanol-treated (TcT), *Chlorella vulgaris* (CvR), *Chlorella vulgaris* ethanol-treated (CvT), *Nannochloropsis gaditana* (NgR), and *Nannochloropsis gaditana* ethanol-treated (NgT).

Bread Ingredients (g/100 g)	Control	TcR	TcT	CvR	CvT	NgR	NgT
Buckwheat flour Provida	46	44.2	44.2	44.2	44.2	44.2	44.2
Rice flourEspiga	31	29.8	29.8	29.8	29.8	29.8	29.8
Potato starchGlobo	23	22.1	22.1	22.1	22.1	22.1	22.1
Microalgae biomass replacement	0	4	4	4	4	4	4
Sunflower oil, Fula (in relation to flour)	5.5	5.5	5.5	5.5	5.5	5.5	5.5
HPMC, Dupont (in relation to flour)	4.6	4.6	4.6	4.6	4.6	4.6	4.6
Instant dried yeast, Fermipan (in relation to flour)	2.8	2.8	2.8	2.8	2.8	2.8	2.8
Commercial salt (in relation to flour)	1.8	1.8	1.8	1.8	1.8	1.8	1.8
Commercial sugar (in relation to flour)	2.8	2.8	2.8	2.8	2.8	2.8	2.8
Macronutrients (algae biomass)							
Proteins (total amino acids)		42.1 ± 0.1	59.5 ± 0.2	47.8 ± 1.1	58.8 ± 0.3	43.3 ± 1.5	61.7 ± 2.8
Lipids		13.8	0.3	15.7	0.6	21.4	0.4
Ash		16.0 ± 0.1	16.7 ± 0.1	6.7 ± 0.6	8.2 ± 0.0	7 ± 1.1	7.4 ± 0.1
Dietary fibers		8.9 ± 0.8	15.1 ± 2.2	13.8 ± 0.5	19.0 ± 0.6	12.2 ± 0.6	21.8 ± 0.8

**Table 2 foods-11-00397-t002:** Mixing properties acquired from Micro-doughLab curves of the control and the microalgae biomass-enriched gluten-free breads: water absorption (WA), dough development time (DDT), and dough stability (DS).

GF Dough	WA (%)	Peak (mN.m)	Torque (600 s)	Torque (1200 s)	DDT (s)	DS (s)	Dough Softening (mN.m)
*Control*	75.0	69.3 ± 2.7	52.5 ^cd^ ± 3.6	50.7 ^cd^ ± 4.1	43.6 ^a^ ± 8.8	28.2 ^a^ ± 4.5	17.8 ^ab^ ± 6.4
*TcR*	74.5	70.0 ± 5.5	48.3 ^d^ ± 1.2	44.3 ^d^ ± 0.5	52.0 ^a^ ± 3.5	22.2 ^a^ ± 9.2	20.0 ^a^ ± 5.9
*TcT*	75.5	70.6 ± 2.3	65.6 ^a^ ± 5.3	63.2 ^a^ ± 2.8	51.0 ^a^ ± 4.2	24.0 ^a^ ± 0.0	6.3 ^c^ ± 1.2
*CvR*	75.5	72.8 ± 2.2	56.2 ^bc^ ± 1.9	56.5 ^bc^ ± 3.2	51.0 ^a^ ± 5.4	26.4 ^a^ ± 3.3	14.0 ^bc^ ± 4.2
*CvT*	75.3	70.4 ± 2.8	57.7 ^bcd^ ± 6.3	54.0 ^bc^ ± 6.1	47.1 ^a^ ± 5.4	24.0 ^a^ ± 0.0	13.6 ^bc^ ± 3.9
*NgR*	75.3	68.7 ± 2.5	51.3 ^cd^ ± 4.6	53.8 ^cd^ ± 3.1	50.2 ^a^ ± 13.3	30.0 ^a^ ± 10.0	16.0 ^bc^ ± 2.8
*NgT*	75.5	72.2 ± 2.2	61.2 ^d^ ± 2.7	58.2 ^ab^ ± 4.1	45.0 ^a^ ± 6.0	24.0 ^a^ ± 0.0	11.0 ^bc^ ± 2.4

*Tetraselmis chuii* (TcR), *Tetraselmis chuii* ethanol-treated (TcT), *Chlorella vulgaris* (CvR), *Chlorella vulgaris* ethanol-treated (CvT), *Nannochloropsis gaditana* (NgR), and *Nannochloropsis gaditana* ethanol-treated (NgT). The given values represent mean ± stdev of at least 3 replicates, while different letters (where appropriate) in the same column indicate a significant difference (*p* > 0.05) using the Fisher LSD test.

**Table 3 foods-11-00397-t003:** Elastic recovery compliance (Je), viscous recovery compliance (Jv), maximum creep compliance (J_max_), zero shear viscosity (η_0_), and parameters from frequency sweep expressed from the power-law model: storage moduli (G’) and flow index (n’), loss moduli (G”), and flow index (n”) of doughs prepared with the control dough and those with 4% replacement of different microalgae biomasses.

GF Dough	Je(1/Pa)	Jv(1/Pa)	Jmax(1/Pa)	η_0_ × 10^−5^ (Pa.s)	G’ = K’ (Pa.s^n’^)	G” = K” (Pa.s^n”^)	n’	n”
*Control*	1.8 ^a^ ± 0.2	2.1 ^a^ ± 0.2	3.8 ^a^ ± 0.4	8.1 ^b^ ± 0.2	30,395 ^b^ ± 2778	11,962 ^abc^ ± 1165	0.23	0.22
*TcR*	1.0 ^c^ ± 0.1	0.9 ^b^ ± 0.1	1.9 ^c^ ± 0.2	13.4 ^a^ ± 0.0	35,659 ^ab^ ± 5222	13,274 ^ab^ ± 1979	0.21	0.23
*TcT*	1.2 ^abc^ ± 0.3	1.1 ^b^ ± 0.5	2.3 ^bc^ ± 0.7	12.4 ^ab^ ± 0.5	33,759 ^b^ ± 5386	11,787 ^bc^ ± 1373	0.22	0.19
*CvR*	1.2 ^abc^ ± 0.1	1.0 ^ab^ ± 0.1	2.1 ^abc^ ± 0.1	12.1 ^ab^ ± 0.1	27,438 ^b^ ± 891	10,212 ^c^ ± 220	0.22	0.21
*CvT*	1.3 ^ab^ ± 0.2	1.4 ^ab^ ± 0.3	2.7 ^ab^ ± 0.5	9.1 ^ab^ ± 0.3	29,544 ^b^ ± 3826	11,331 ^bc^ ± 929	0.23	0.21
*NgR*	1.1 ^bc^ ± 0.1	1.0 ^ab^ ± 0.2	2.1 ^bc^ ± 0.2	12.5 ^ab^ ± 0.4	35,239 ^ab^ ± 8464	12,769 ^abc^ ± 2979	0.23	0.23
*NgT*	1.2 ^bc^ ± 0.5	1.2 ^ab^ ± 0.7	2.4 ^bc^ ± 1.1	11.1 ^ab^ ± 0.4	39,945 ^a^ ± 6624	14,373 ^a^ ± 2430	0.22	0.20

*Tetraselmis chuii* (TcR), *Tetraselmis chuii* ethanol-treated (TcT), *Chlorella vulgaris* (CvR), *Chlorella vulgaris* ethanol-treated (CvT), *Nannochloropsis gaditana* (NgR), and *Nannochloropsis gaditana* ethanol-treated (NgT). The given values represent mean ± stdev (where mentioned) of at least 3 replicates, while different letters in the same column indicate a significant difference (*p* > 0.05) using the Fisher LSD test.

**Table 4 foods-11-00397-t004:** Major chemical composition (g/100 g) and gross energy value of the control and microalgae biomass-enriched gluten-free breads. *Tetraselmis chuii* (TcR), *Tetraselmis chuii* ethanol-treated (TcT), *Chlorella vulgaris* (CvR), *Chlorella vulgaris* ethanol-treated (CvT), *Nannochloropsis gaditana* (NgR), and *Nannochloropsis gaditana* ethanol-treated (NgT).

GF Bread	Moisture	Protein (N× 6.25)	Fat	Ash	* Carbohydrate	Energy Kcal/100 g
Control	44.96 ^a^ ± 1.17	6.28 ^e^ ± 0.04	3.63 ^a^ ± 0.58	1.47 ^c^ ± 0.00	43.66	232.19
TcR	43.18 ^a^ ± 1.96	7.29 ^c^ ± 0.04	4.00 ^a^ ± 0.35	1.81 ^a^ ± 0.03	42.95	236.96
TcT	43.91 ^a^ ± 2.21	7.35 ^bc^ ± 0.10	3.59 ^a^ ± 0.53	1.80 ^a^ ± 0.00	43.35	235.11
CvR	44.41 ^a^ ± 1.93	7.29 ^c^ ± 0.04	4.02 ^a^ ± 0.35	1.60 ^bc^ ± 0.05	42.68	236.06
CvT	43.04 ^a^ ± 0.80	7.87 ^a^ ± 0.06	3.59 ^a^ ± 0.73	1.65 ^b^ ± 0.00	44.69	239.19
NgR	43.60 ^a^ ± 1.25	7.16 ^d^ ± 0.07	4.55 ^a^ ± 0.17	1.62 ^b^ ± 0.04	43.07	241.87
NgT	44.51 ^a^ ± 1.69	7.44 ^b^ ± 0.01	3.96 ^a^ ± 0.57	1.61 ^bc^ ± 0.02	42.49	235.36

* Carbohydrates values were estimated by differences. Values represent the means (n = 3, wet basis) while different letters in the same column indicate a significant difference (*p* > 0.05) using Fisher’s LSD test.

**Table 5 foods-11-00397-t005:** Influence of 4% microalgae replacement on mineral content (mg/100 g) of the gluten-free bread (n = 2, wet basis). Tetraselmis chuii (TcR), Tetraselmis chuii ethanol-treated (TcT), Chlorella vulgaris (CvR), Chlorella vulgaris ethanol-treated (CvT), Nannochloropsis gaditana (NgR), and Nannochloropsis gaditana ethanol-treated (NgT).

GF Bread	K	Ca	Mg	P	S	Fe	Cu	Zn	Mn
Control	199.2 ± 2.7	6.4 ± 0.1	56.4 ± 0.6	140.0 ± 1.6	70.0 ± 0.2	2.6 ± 0.1	0.1 ± 0.0	0.9 ± 0.0	0.5 ± 0.0
TcR	223.7 ± 1.8	67.5 ± 1.6	68.6 ± 1.2	164.6 ± 0.3	104.3 ± 0.4	4.8 ± 0.4	0.2 ± 0.0	1.0 ± 0.0	0.7 ± 0.0
TcT	201.6 ± 0.5	88.7 ± 0.3	64.1 ± 0.1	173.3 ± 0.4	103.8 ± 0.8	5.5 ± 0.2	0.1 ± 0.0	1.0 ± 0.0	0.8 ± 0.0
CvR	218.6 ± 1.2	8.0 ± 0.2	62.6 ± 0.1	166.5 ± 1.3	94.7 ± 0.3	2.9 ± 0.1	0.2 ± 0.0	0.9 ± 0.0	0.6 ± 0.0
CvT	220.4 ± 2.2	9.2 ± 0.2	69.8 ± 0.8	183.4 ± 2.7	108.7 ± 1.0	3.8 ± 0.2	0.2 ± 0.0	1.0 ± 0.0	0.7 ± 0.0
NgR	202.4 ± 2.8	10.1 ± 0.1	63.1 ± 0.2	162.1 ± 2.6	94.0 ± 2.0	3.3 ± 0.0	0.1 ± 0.0	1.0 ± 0.0	0.6 ± 0.0
NgT	202.3 ± 3.2	14.0 ± 0.1	68.7 ± 0.4	173.8 ± 0.6	99.8 ± 0.4	2.8 ± 0.0	0.1 ± 0.0	1.1 ± 0.0	0.7 ± 0.0
15% RDV * (mg)	300.0	120.0	56.3	105.0	NM	2.1	0.2	1.5	0.3

* Recommended daily value (RDV) per European Community Regulation N,1924/2006, Directive N-9090/494 (CE). NM not mentioned.

**Table 6 foods-11-00397-t006:** Bread crumb and crust color parameters. *Tetraselmis chuii* (TcR), *Tetraselmis chuii* ethanol-treated (TcT), *Chlorella vulgaris* (CvR), *Chlorella vulgaris* ethanol-treated (CvT), *Nannochloropsis gaditana* (NgR), and *Nannochloropsis gaditana* ethanol-treated (NgT).

	Crumb				Crust			
GF Bread	L*	a*	b*	∆E	L*	a*	b*	∆E
Control	67.53 ^a^ ± 1.8	2.46 ^ab^ ± 0.2	12.08 ^a^ ± 1.3		49.93 ^a^ ± 2.6	11.10 ^a^ ± 0.8	19.78 ^a^ ± 2.7	
TcR	34.99 ^d^ ± 3.6	−0.23 ^ab^ ± 0.2	10.24 ^cd^ ± 1.9	32.7	36.32 ^b^ ± 4.5	1.07 ^c^ ± 0.8	8.52 ^cd^ ± 2.7	20.3
TcT	51.07 ^c^ ± 3.7	0.05 ^ab^ ± 0.0	11.40 ^c^ ± 1.5	16.6	41.62 ^ab^ ± 4.5	5.56 ^b^ ± 0.5	12.74 ^abcd^ ± 2.5	12.2
CvR	35.75 ^d^ ± 4.7	−0.75 ^b^ ± 0.1	8.83 ^b^ ± 2.5	32.1	35.54 ^b^ ± 3.3	0.71 ^c^ ± 0.5	6.08 ^d^ ± 1.0	22.4
CvT	52.28 ^bc^ ± 2.9	1.71 ^a^ ± 0.4	15.21 ^a^ ± 2.4	15.6	46.19 ^ab^ ± 5.5	5.99 ^b^ ± 0.4	15.74 ^abc^ ± 3.1	7.5
NgR	37.22 ^d^ ± 4.1	−0.60 ^ab^ ± 0.3	14.05 ^ab^ ± 3.0	30.5	35.51 ^b^ ± 5.7	1.40 ^c^ ± 0.1	9.47 ^bcd^ ± 3.3	20.2
NgT	60.99 ^ab^ ± 4.7	2.70 ^a^ ± 0.3	14.83 ^a^ ± 1.8	7.1	47.15 ^ab^ ± 6.1	9.28 ^a^ ± 1.0	17.76 ^ab^ ± 3.8	3.9

Different letters within the same column indicate that the values are significantly different at *p* < 0.05 using the Fisher LSD test.

## Data Availability

The data presented in this study are available on request from the corresponding author.

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
