# Peer review of "Improving the Nutritional, Structural, and Sensory Properties of Gluten-Free Bread with Different Species of Microalgae"

_foods, 2022, doi:10.3390/foods11030397_

Round 1

Reviewer 1 Report

- line 47:delate the (GF) because there is written in the text, -line 59: delate that - line 78: substitute The with the - line 93: define HPMC - lines 97-101 should be put before lie 91 - line 101: in the reference 24 the authors use the term Microchloropsis instead of Manochloropsis. You shoud explain in the text that they are synonyms - Table 1: there is the chemical composition of the biomass. Maybe it is better to create a separate table - line 124: missed the +/- - line 132: G'' loss modules . line 140: indicate the meaning of the symbols used - line 159: delate flour - line 219: substitute The with the - Table 2: indicate the meaning of DDT and DS - Table 4 and 5: in the text (lines 153-171) the authors spoke of protein, ask and so on measured on the dry product. Why the data are express as g/100g of product? - line 384: before [22] delate ( - Figure 5: indicate the unit used to express the DPPH - line 422: are you sure about deltaE>5 usually is 2.5? - line 495 substitute Ethanol with ethanol -

Reviewer 2 Report

After reading the manuscript " Improving the nutritional, structural and sensory properties of gluten-free bread with marine microalgae: Tetraselmis chuii, Chlorella vulgaris and Nannochloropsis gaditana", I realized that the manuscript showed in some parts the scientific rigour wanted, but in others parts I have missed it.The references are not exactly current, besides the title and objective could be more attractive and cientific.Thats why i have written some suggestions in an attempt to improve them.

L.1- The title is too extensive, what do you think about removing the specifications on algae from the title, they will be presented in the Material and methods.

L.35-L.36- It is not only people with celiac disease who cannot consume gluten, you could also include diseases associated with gluten: non-celiac gluten sensitivity, wheat allergy, gluten ataxia and dermatitis herpetiformis.

L.44- set GF the 1st time it appears in the Introduction, please. L.48 was not the first time.

L.66 I think it is relevant to define the acronym DW in the 1st quotation.

L.66- "namely calcium phosphorus"- Please, check the punctuation marks here.

L.77- include this part :"Microalgae acquire their dark green colour from pigments (chlorophylls)", immediately after "dark green colour" L.75.              I have the feeling that the paragraph will flow more effectively.

L.78- 88 -  Please, evaluate the relevance of everything that is written, it has become too long and confusing.

Maybe something like this: The aim of this paper was to evaluate nutritional and  technological  quality of  GF bread recipes  enriched with microalgae besides  thsensory attributes.

L.93- I think it is important to explain the importance of the use and what  the HPMC is, because some people have never used it.

 L. 104- Table 1-Please, give more details about: sunflower(flour? oil ?), What kind of sugar? Salt ? Dried yeast ( instant ? active?)

L.106- I think it was not appropriate to present results in the Material and methods.

L.179- I did not find Mohankumar (2018) in your references, please check  it.

L.191-  Please, we have protocols to perform sensory analysis. You need to insert  more details about how the test was performed. Which test ? 33 assessors, but how many men and women? Which author/year did you follow for only 33 assessors?  booths ?  Were the ethical principles of the Helsinki Declaration followed for the sensory analysis ? This section needs to be more detailed, very important pieces of information were missing.

L.226- Please reduce the thickness of the lines in Figure 1, we can barely see the control treatment and the CvR

L.231- Table 2 can be improved, it looks like a chart, not a table, insert the information above the "control" cell.
Please define the acronyms: WA,DDT (s) and DS. Wouldn't it be interesting to present if there was a statistical difference from Peak as well?
Wouldn't it be useful, through the rounding rule, to clean up this table a little bit?

L.262- Figure 2a- enlarge the legend a little, please.

L.321- I think the result for specific volume of the "control" needs to be discussed more widely. Specific volume is an important parameter to analyze the quality of bread which involves loaf volume and loaf weight. We can have an interesting result here in this association that you have proposed.

L.333- Reorganise figure 3 into a and b, as you did with figure 2, it looks better.

L.350- PCA analysis can explain 64%, do you guys think this could be a bias in the paper? 

L.466- Padronize color and flavor (Figure 7 a) x colour and flavour in the text

L. 470- Global appreciation (Figure 7b)- I really didn't understand what the authors wanted to express here. In L.194, you use  global appreciate. What about general apperance ? Please, correct, I think something is wrong.

L.472- In figure 7b wasn't the objective to compare the treatments (control, CVR, CVT)? The figure is not giving this impression. Please, think about it.

L. 467/ 472 - By the way, Buying intent (Figure 7b) ou buying intention, check, please.

L.490- In the conclusion "ethanol" was emphasized in the paper, but I had not understood it as an objective. On the other hand, in the paragraph before the objective you mentioned "ethanol", I would include it in the Material and methods.  While, other important results were not addressed in the conclusion. Please think about it.

Reviewer 3 Report

It is very important to find alternative food sources, which are sustainable and also of nutritional benefit to people. Microalgae are important ingredients because they are rich in both protein and bioactive compounds. Taking the above into consideration the work "Improving the nutritional, structural and sensory properties of gluten-free bread with marine microalgae: Tetraselmis chuii, Chlorella vulgaris and Nannochloropsis gaditana"  in my opinion is interesting. Reviewer suggest  some adjustments (deleting double spaces: for example: Line 124, Line 163, Line 166, Line 169. Line 224; insertion of space: for example: Line 15, Line 48, Line 119, Line156, Line 180, Line 181, Line 210, Line 317, Line 339; deletion of initial bracket: Line 357, Line 384). This study should be of interest to the readers of Special Issue of Foods (The Application of Microalgae for the Development of High-Added-Value Products). 

Round 2

Reviewer 2 Report

After another evaluation of the manuscript, I see a great improvement in the quality of the paper. The authors have accepted almost all of my requests, some they declined and justified.   BUT,  1- Mohankumar is the second author, not the first as you cited on line 189. Correct, please. 2- The figures still have difference in the size of the legend in Figure 2B, please improve it 3- Buying intention L. 211, correct please.  

Author Response

Thanks for your valuable comments, which improved our work a great deal. Below is response to your further comments

1- Mohankumar is the second author, not the first as you cited on line 189. Correct, please. 

Ans: Mohankumar is actually the 1st author, not the 2nd. The long name of author might be confusing. 

Please see link to the attached paper.  http://jst.org.in/wp-content/uploads/2020/07/16.-Total-Phenolic-Content-of-Organic-and-Conventionally-Grown-Gourd-Vegetables.pdf

2- The figures still have difference in the size of the legend in Figure 2B, please improve it

Ans: Figures improved (legends size and x, y co-ordinates of Fig. 2B made more consistent)

3- Buying intention L. 211, correct please.  

Ans: Corrected as requested in the text.
